# Strategic Fluorination to Achieve a Potent, Selective, Metabolically Stable, and Orally Bioavailable Inhibitor of CSNK2

**DOI:** 10.3390/molecules29174158

**Published:** 2024-09-02

**Authors:** Han Wee Ong, Xuan Yang, Jeffery L. Smith, Sharon Taft-Benz, Stefanie Howell, Rebekah J. Dickmander, Tammy M. Havener, Marcia K. Sanders, Jason W. Brown, Rafael M. Couñago, Edcon Chang, Andreas Krämer, Nathaniel J. Moorman, Mark Heise, Alison D. Axtman, David H. Drewry, Timothy M. Willson

**Affiliations:** 1Rapidly Emerging Antiviral Drug Development Initiative (READDI), Chapel Hill, NC 27599, USAtim.willson@unc.edu (T.M.W.); 2Structural Genomics Consortium (SGC) and Division of Chemical Biology and Medicinal Chemistry, Eshelman School of Pharmacy, University of North Carolina at Chapel Hill, Chapel Hill, NC 27599, USA; 3Department of Genetics, University of North Carolina at Chapel Hill, Chapel Hill, NC 27599, USA; 4Department of Microbiology & Immunology, University of North Carolina at Chapel Hill, Chapel Hill, NC 27599, USA; 5Lineberger Comprehensive Cancer Center, University of North Carolina at Chapel Hill, Chapel Hill, NC 27599, USA; 6Department of Chemistry, University of North Carolina at Chapel Hill, Chapel Hill, NC 27599, USA; 7Takeda Development Center Americas, Inc., San Diego, CA 92121, USA; 8Centro de Química Medicinal (CQMED), Centro de Biologia Molecular e Engenharia Genética (CBMEG), University of Campinas, Campinas 13083-886, SP, Brazil; 9Structural Genomics Consortium (SGC), Institute of Pharmaceutical Chemistry, Goethe University Frankfurt am Main, Max-von-Laue-Str. 9, 60438 Frankfurt am Main, Germany

**Keywords:** CSNK2, pyrazolo[1,5-*a*]pyrimidine, fluorination, antiviral, SARS-CoV-2, β-coronavirus

## Abstract

The host kinase casein kinase 2 (CSNK2) has been proposed to be an antiviral target against β-coronaviral infection. To pharmacologically validate CSNK2 as a drug target in vivo, potent and selective CSNK2 inhibitors with good pharmacokinetic properties are required. Inhibitors based on the pyrazolo[1,5-*a*]pyrimidine scaffold possess outstanding potency and selectivity for CSNK2, but bioavailability and metabolic stability are often challenging. By strategically installing a fluorine atom on an electron-rich phenyl ring of a previously characterized inhibitor **1**, we discovered compound **2** as a promising lead compound with improved in vivo metabolic stability. Compound **2** maintained excellent cellular potency against CSNK2, submicromolar antiviral potency, and favorable solubility, and was remarkably selective for CSNK2 when screened against 192 kinases across the human kinome. We additionally present a co-crystal structure to support its on-target binding mode. In vivo, compound **2** was orally bioavailable, and demonstrated modest and transient inhibition of CSNK2, although antiviral activity was not observed, possibly attributed to its lack of prolonged CSNK2 inhibition.

## 1. Introduction

Since its emergence in 2019, the β-coronavirus severe acute respiratory syndrome coronavirus-2 (SARS-CoV-2) has caused over 775 million infections and over seven million deaths worldwide [1]. Importantly, it is only the latest β-coronavirus to cause a pandemic; there have been previous examples of β-coronavirus-induced outbreaks, such as the original SARS outbreak in 2003 and the MERS outbreak in 2012, and yet other members of the β-coronavirus family are primed for emergence to cause future pandemics [2,3]. This demonstrates the need for effective antivirals against β-coronaviruses.

Most antivirals target virus-encoded factors such as viral proteases and polymerases and are termed direct-acting antivirals [4,5,6,7]. These include the FDA-approved drugs and drugs under emergency use authorization against SARS-CoV-2: nirmatrelvir, ritonavir, remdesivir, and molnupiravir [8]. Host-directed therapy is a promising alternative for developing novel antivirals. In contrast to direct-acting antivirals, host-directed therapeutic agents target host factors hijacked by the virus for its life cycle. This mechanism has the advantages of reduced propensity for resistance development, and the potential for broad-spectrum activity against multiple viruses in the same family [9,10,11]. Host-directed therapeutic agents thus have the potential to safeguard humanity against β-coronaviruses that emerge in the future. One of the key challenges in host-directed therapy is the identification and validation of a suitable antiviral target. One such promising target is the host kinase casein kinase 2 (CSNK2, also known as CK2), which is hijacked by β-coronaviruses for their replication cycle [12,13,14,15]. We and others have previously shown that CSNK2 inhibitors are able to inhibit β-coronaviral replication in vitro [13,16,17,18,19,20], and it is our long-term goal to validate CSNK2 as an antiviral target in vivo.

One chemotype extensively investigated for CSNK2 inhibition is the pyrazolo[1,5-*a*]pyrimidine scaffold [20,21,22,23,24,25]. Unlike the widely used CSNK2 inhibitor silmitasertib, which has multiple kinase off-targets at its commonly used doses, limiting its utility as a pharmacological probe [24], the pyrazolo[1,5-*a*]pyrimidines have demonstrated exquisite kinome-wide selectivity and are highly valued as in vitro CSNK2 chemical probes [24,26]. In previous efforts towards identifying a potent yet selective CSNK2 inhibitor for in vivo applications, we identified both phase I and phase II metabolism, particularly GSH conjugation, to be a major metabolic transformation of this scaffold in mouse hepatocytes in vitro and in mice in vivo [20]. In a metabolite identification study conducted for compound **1** (see Table 1 for structure), GSH conjugation was one of the primary metabolic pathways [20]. Given the excellent potency and aqueous solubility of compound **1**, we sought to make minor modifications to its structure that could address this metabolic liability while maintaining its favorable pharmacological properties. In this work, we report the discovery of compound **2** through the strategic fluorination of compound **1**, improving upon its in vivo pharmacokinetic (PK) profiles while maintaining other favorable properties of CSNK2 and antiviral potency, kinome-wide selectivity, solubility, and low cytotoxicity. Additionally, we present a co-crystal structure to support the on-target binding mode of compound **2** with CSNK2A1. Despite improvements in the PK profile, only a transient and modest inhibition of CSNK2 catalytic activity was observed in vivo, which was insufficient for pharmacological suppression of viral replication. Nevertheless, herein we disclose compound **2** as a potent, selective, and orally active CSNK2 inhibitor with a well-characterized PK profile.

## 2. Results

### 2.1. Discovery of Compound **2** as a Potent and Selective CSNK2 Inhibitor

Building on the metabolic identification of compound **1** [20], we hypothesized that oxidation by CYP450 enzymes occurred at the electron-rich trisubstituted phenyl ring, generating an arene oxide or quinone-imine that then enabled subsequent GSH conjugation [27]. Fluorination is a well-established strategy for reducing electron density and improving the stability of phenyl and heterocyclic rings to oxidation [28]. We thus introduced a fluorine atom at the *ortho*-position of the aniline ring (compound **2**, Table 1). This modification retained the low nanomolar potency of compound **1** against both CSNK2A1 and CSNK2A2 as measured using the NanoBRET assay [16,20,24], as well as the submicromolar antiviral potency against the β-coronaviruses mouse hepatitis virus (MHV) and SARS-CoV-2. The minor increase in lipophilicity of the hydrogen-to-fluorine substitution did not significantly impact physiochemical properties, and compound **2** remained highly soluble, with an aqueous solubility of 59 μg/mL. The cytotoxicity of compound **2** was also similar to that of compound **1** in A549-ACE2 cells, with no toxicity at 0.1 μM and low toxicity at 1.0 μM. To demonstrate CSNK2 inhibition using an orthogonal method, we treated A549-ACE2 cells with 1 μM and 5 μM of compound **2** for 24 h. A dose-dependent reduction in the phosphorylation levels of the CSNK2 substrate EIF2S2 [29] was observed (Appendix A), indicating that compound **2** inhibited downstream CSN2K2 signaling in a cellular context. Compound **2** is thus a potent CSNK2 inhibitor with favorable properties for further compound progression.

**Table 1 molecules-29-04158-t001:**
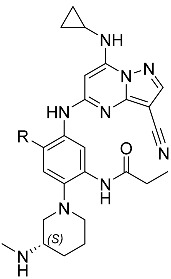
Effect of fluorination on the phenyl ring.

Compound	R	CSNK2A1 pIC_50_ ^a^	CSNK2A2 pIC_50_ ^a^	MHV pIC_50_ ^b^	SARS-CoV-2 pIC_50_ ^c^	Kinetic Solubility (μg/mL) ^d^	% Viability of A549-ACE2 at 0.1 μM ^e^	% Viability of A549-ACE2 at 1.0 μM ^e^
**1**	H	7.6 ^f^	7.7 ^f^	6.9 ^f^	6.2	71 ^f^	100	79
**2**	F	8.1	7.9	6.8	6.5	59	100	76

^a^ Performed in singlicate. ^b^ Mean of three independent experiments performed in triplicate. ^c^ Performed in triplicate. ^d^ Kinetic solubility measurements were carried out in phosphate-buffered saline solution (PBS) at pH 7.4 from DMSO stock solutions. ^e^ Cytotoxicity measurements using the CellTiter-Glo assay performed in quadruplicate. ^f^ Data for compound **1** from ref. [20].

To characterize the kinome-wide selectivity profile, we screened compound **2** at 10 μM against 192 human kinases in the NanoBRET K192 panel [30] (Figure 1). Only 8 out of the 192 kinases investigated had >50% occupancy of compound **2**, including CSNK2A1 and CSNK2A2 (Table 2 and Appendix A). A follow-up dose–response experiment was then conducted using the NanoBRET assay for the off-targets identified in the single-concentration experiment. The IC_50_ of compound **2** against CLK2, CLK4, DAPK2, and PHKG1 were above the highest concentration tested in the dose–response experiment (30 μM), while CLK1 was inhibited with an IC_50_ of 29 μM. Due to a poor assay window, an IC_50_ for DYRK1A was not conclusively determined, although no occupancy was observed up to 10 μM in the dose–response experiment. Our results thus demonstrate that compound **2** is an exquisitely selective CSNK2 inhibitor.

We next co-crystalized compound **2** with CSNK2A1 to understand its binding mode (Figure 2A, see Appendix A for crystallographic refinement statistics). Compound **2** bound in a similar binding mode to the previously reported CSNK2 inhibitors of the pyrazolo[1,5-*a*]pyrimidine scaffold [21,22,23,24,25]. The N1 of the pyrazolo[1,5-*a*]pyrimidine core and the exocyclic NH at the 7-position of the pyrazolo[1,5-*a*]pyrimidine of compound **2** interacted with the hinge region of the kinase, forming hydrogen bonds with the hinge residue Val116. The nitrile at the 3-position of the pyrazolo[1,5-*a*]pyrimidine similarly formed a hydrogen bond with a water molecule buried in the ATP-binding pocket. This water molecule also formed a hydrogen bond with the carbonyl group of the amide. As far as we know, this was also the first time a crystal structure of an inhibitor of this series with a 3-aminopiperidine at the *para*-position of the aniline ring was reported. While acyclic amines at this position point towards the C-terminal lobe to interact with Asn161 [23,25], here, the amino group on the piperidine was instead directed towards the N-terminal lobe, forming a hydrogen bond with Ser51 on the P-loop. Compared to the co-crystal structure of the CSNK2 chemical probe **SGC-CK2-1** with CSNK2A1 [24] (Figure 2B), the sterically demanding piperidine ring of compound **2** induces a slight leftward shift of the aniline ring towards the solvent, but the overall alignment between these two structures is excellent.

### 2.2. Pharmacokinetic Characterization

Next, we advanced compound **2** into pharmacokinetic experiments to assess the impact of the additional fluorine atom on metabolic clearance. When profiled in vitro, the clearance rates of compound **2** by mouse and human liver microsomes and hepatocytes were similar to those of compound **1** (Appendix A). However, in vivo, when administered to mice via i.p. injection in a 24 h PK study, compound **2** demonstrated an improved half-life (2.5 h) and AUC (10,100 h × nM) compared to compound **1** (t_1/2_ = 1.2 h, AUC = 7010 h × nM) when administered via i.p. injection [20] (Table 3). The i.v. half-life and clearance of both compounds were, relative to the i.p. results, comparatively similar to each other. These results suggested that, through the installation of a key fluorine atom, we were able to reduce the first-pass metabolism and improve compound exposure via the i.p. route of administration of compound **2**.

In addition, when compound **2** was administered orally to mice at 10 mg/kg, we were pleasantly surprised that it demonstrated good oral bioavailability (F = 52%) (Table 3, Figure 3). The oral bioavailability of compound **2** was an improvement over a recently published series of pyrazolo[1,5-*a*]pyrimidine inhibitors with a triazole bioisostere of the amide [25] and related pyrazolo[1,5-*a*]pyrimidine inhibitors disclosed by AstraZeneca [21,22,23]. As these pyrazolo[1,5-*a*]pyrimidines have previously been reported to accumulate in the lungs [25], we similarly evaluated the lung concentrations of compound **2** when administered to mice. Lung accumulation was of particular interest to us as the lungs are a key organ for SARS-CoV-2 infection. Indeed, we found that compound **2** also accumulated in the lungs when measured 12 and 24 h post dose administration.

To improve exposure of related CSNK2 inhibitors of the pyrazolo[1,5-*a*]pyrimidine series, we previously co-dosed compounds in vivo with the CYP450 inhibitor 1-ABT [32] and glutathione-*S*-transferase (GST) inhibitor EA [33], and successfully improved the half-life and AUC of a previous lead compound [20]. We thus similarly applied this strategy to compound **2** here. We dosed mice with compound **2** i.p. at 10 mg/kg in a snapshot PK study (Figure 4). Using the previously established protocol, one cohort of mice was also co-dosed and pre-treated (6 h) with EA (30 mg/kg i.p.), and pre-treated (2 h) with 1-ABT (100 mg/kg p.o.), while the other cohort had no co-treatments. Our results revealed that there were no significant differences between the plasma and lung concentrations of the two cohorts of mice, and we achieved a similar half-life and AUC for both cohorts (Table 4). These co-dosing experiments further demonstrate that through strategic fluorination, we reduced metabolic liability such that CYP450- and GST-mediated metabolism was no longer the primary cause of clearance of compound **2** in vivo.

We conducted a multi-dose PK study of compound **2** in mice, administering compound **2** orally at a dose of 30 mg/kg every 12 h for 48 h (Figure 5). Across multiple doses, we observed similar C_peak_ and C_trough_ values, indicating that compound **2** undergoes regular metabolism, with no induction of metabolism or in vivo accumulation over multiple doses. We noted that the total plasma level of compound **2** when administered at 30 mg/kg p.o. twice daily (b.i.d.) was usually above the concentration required to inhibit SARS-CoV-2 in vitro, and the C_trough_ was close to falling below the IC_50_, although only the C_peak_ levels were close to the IC_90_. Combined with the lung accumulation of compound **2** at the 48 h time point, we initially hypothesized that this dose might be sufficient to maintain antiviral activity in vivo.

### 2.3. In Vivo Efficacy

For the in vivo SARS-CoV-2 antiviral efficacy study, we administered compound **2** at 30 mg/kg p.o. to mice every 12 h. Treatment was started 12 h before viral inoculation and viral titer was measured in mouse lungs 24 h post-infection (36 h after first compound administration). Regrettably, we did not observe any effect on viral titer in the mice following treatment (Figure 6A).

In parallel, to demonstrate in vivo target engagement of CSNK2, we dosed mice with compound **2** using the same dosing regimen without viral inoculation. Mouse lung samples were harvested 3, 6, and 36 h after the first dose, and phosphorylation levels of the CSNK2 substrates AKT [34] and EIF2S2 [29] were measured using Western blot. A decrease in the phosphorylation levels of Ser129 of AKT, which was statistically significant (*p* = 0.0028) based on an unpaired *t*-test with Welch’s correction, was observed at the 3 h time point that corresponds to the plasma t_max_ of compound **2** when administered orally (Figure 6B). Although no statistically significant difference in phosphorylation of Ser2 of EIF2S2 was observed, there was a small decrease at the 3 h time point that did not reach statistical significance (*p* = 0.14) (Figure 6C). These results provide an explanation for the lack of inhibition of viral replication. It was likely that sustained inhibition of CSNK2 catalytic activity required for disruption of key viral replication processes was not achieved throughout the time course of the in vivo antiviral study.

### 2.4. Chemistry

The synthesis of compound **1** was reported previously [20]. Compound **2** was synthesized in a convergent manner as described in Figure 1. The 7-chloro substituent on 5,7-dichloropyrazolo[1,5-*a*]pyrimidine-3-carbonitrile (**3**) was displaced with cyclopropylamine to obtain compound **4**. In parallel, an S_N_Ar reaction between compounds **5** and **6** selectively displaced the fluoro group *ortho*- to the nitro group (**7**). The regioselectivity of the reaction was confirmed by ^1^H-^19^F HOESY (Appendix A). The aniline was next protected by the Cbz protecting group (**8**), and the nitro group was subsequently reduced by iron to form an aniline (**9**) and acylated with propanoyl chloride (**10**). The Cbz group of the aniline was removed via hydrogenation (**11**), and the deprotected aniline subsequently underwent a Buchwald–Hartwig coupling with compound **4** to produce compound **12**, which was finally treated with TFA to remove the Boc group, yielding compound **2**.

## 3. Discussion

The pyrazolo[1,5-*a*]pyrimidines are a highly potent series of CSNK2 inhibitors with improved selectivity compared to other ATP-competitive inhibitors [24]. However, the application of this scaffold to the development of CSNK2 inhibitors with in vivo activity has been met with challenges due to its physiochemical and PK properties [20,21,22,23]. For example, **SGC-CK2-1**, a high-quality chemical probe with proven utility in cellular studies of CSNK2 [24,26], has poor solubility and poor stability in mouse liver microsomes [20]. In another example, we recently reported a series of 1,2,4-triazole-containing compounds of this chemotype that demonstrated a balance of potency, solubility, and metabolic stability, but was not orally bioavailable in mice [25]. To improve the solubility and microsomal stability of **SGC-CK2-1**, we installed amines at the *para-*position of the phenyl ring, only to be met with challenges in hepatocyte stability and high in vivo clearance in mice attributed to phase I and phase II metabolism [20].

Compound **2** was developed by the installation of a fluorine atom at an appropriate position in compound **1**, an exemplar of the pyrazolo[1,5-*a*]pyrimidine series. Compound **2** was found to be highly potent against CSNK2 in the cellular context and demonstrated antiviral activity against both MHV and SARS-CoV-2 in vitro, with submicromolar IC_50_s. When screened against 192 kinases across the human kinome, compound **2** was exquisitely selective for CSNK2A1 and CSNK2A2, with only six other kinases with >50% occupancy at 10 μM. Of the six off-target kinases identified in the single-point screen, inhibition was only observed in the mid-micromolar range in dose–response experiments, offering at least a 1000-fold selectivity compared to CSNK2A1 and CSNK2A2. Furthermore, a crystal structure obtained with CSNK2A1 further supports its on-target binding mode. We envision that compound **2** may serve as a useful molecule for facilitating the investigation of CSNK2 function.

Importantly, this hydrogen-to-fluorine substitution improved the in vivo metabolic stability of compound **1** in mice. Compound **2** is orally bioavailable, which demonstrates an improvement over our previously reported lead CSNK2 pyrazolo[1,5-*a*]pyrimidine inhibitors with 1,2,4-triazole as an amide bioisostere [25] and has improved bioavailability compared to analogs reported by AstraZeneca [21,22,23]. Like the lead compound with a 1,2,4-triazole, compound **2** also showed higher lung concentrations than plasma concentrations, suggesting that lung accumulation was not a property unique to the 1,2,4-triazole but may also be extended to other pyrazolo[1,5-*a*]pyrimidine inhibitors. Through co-dosing studies with EA and 1-ABT, we showed that CYP450- and GST-mediated metabolism was no longer a primary cause of in vivo clearance of compound **2**. Our efforts thus demonstrate a successful strategy for mitigating the rapid in vivo clearance of the pyrazolo[1,5-*a*]pyrimidines caused by GSH conjugation. The learning points from our strategic fluorination methodology may be further extended to other chemotypes.

When evaluated for efficacy in vivo, we found that despite maintaining a plasma concentration at or above the SARS-CoV-2 IC_50_, we failed to achieve sustained inhibition of CSNK2 catalytic activity or SARS-CoV-2 replication in vivo. We thus followed up with investigations of the free plasma concentration of compound **2**. Prior studies have suggested that the in vivo free drug concentration may need to remain above the EC_90_ levels determined in cellular assays for therapeutic effect [35,36]. This suggests that the free drug concentration of compound **2** was thus likely insufficient at the 30 mg/kg dose since the projected minimum concentration based on the SARS-CoV-2 IC_90_ was 2.2 μM, 12-to-19-fold higher than the free plasma C_peak_ concentration (120–190 nM, Figure 5) considering the unbound fraction in mouse plasma (f_u_) of 0.0605 for compound **2**. As a biomarker for on-target efficacy, we also measured the phosphorylation levels of CSNK2 substrates EIF2S2 and AKT in mice treated with the same dosing regimen. A modest decrease in AKT phosphorylation was only observed near the t_max_ of compound **2** when dosed orally. The free plasma concentration of **2** was above the CSNK2 NanoBRET IC_50_ for the entirety of the 48 h dosing period (Figure 5). However, since the NanoBRET IC_90_ was only reached at the C_peak_ concentrations (2, 14, 26, and 38 h time points), CSNK2 inhibition would only be achieved transiently during the in vivo antiviral study. It seems that, while we mitigated CYP450-mediated metabolism in vivo, it is possible that there were other modes of clearance for the compound that are still present that led to insufficient concentrations for sustained inhibition of CSNK2. While transient inhibition of CSNK2 was indeed achieved, the lack of sustained inhibition throughout the whole time course of the antiviral study was likely the reason for the lack of efficacy against SARS-CoV-2 replication. Further optimization of the dosing regimen will be necessary to achieve the sustained inhibition of CSNK2 catalytic activity by compound **2** in vivo demanded by the antiviral study. However, compound **2** may still be a useful pharmacological tool for disease indications that only require transient inhibition of CSNK2 signaling [37].

In conclusion, we have identified pyrazolo[1,5-*a*]pyrimidine **2** as a potent, selective, and cell-active CSNK2 inhibitor. Strategic fluorination of the pyrazolo[1,5-*a*]pyrimidine chemotype proved to be a productive approach for reducing in vivo metabolism and improving the PK profile of our lead compound. We envisage that this strategy could be further applied to the future development of pyrazolo[1,5-*a*]pyrimidine inhibitors and beyond. Compound **2** is the pyrazolo[1,5-*a*]pyrimidine inhibitor with the best in vivo oral bioavailability reported to date, and may be used for the investigation of CSNK2 function in disease models where transient inhibition is sufficient for therapeutic efficacy.

## 4. Materials and Methods

### 4.1. Synthesis of Compound **2**

All chemical reagents were commercially available except those whose syntheses are described below. All reaction mixtures and column eluents were monitored via analytical thin-layer chromatography (TLC) performed on precoated fluorescent silica gel plates, 200 μm with an F254 indicator; visualization was accomplished by UV light (254/365 nm). LC-MS measurements were determined on Shimadzu LC-AB + LCMS-2020, Shimadzu LC-AD + LCMS-2020 (Kyoto, Japan), Shimadzu LC-AD xR + LCMS-2020, or Agilent 1200 (Santa Clara, CA, USA) + Infinitylab LC/MSD instruments (Castle Rock, CO, USA). Purity was determined by HPLC measurement using a Shimadzu LC-20 + LCMS-2020 instrument fitted with an Agilent PoroShell 120 EC-C18 column (45 °C, 2.7 μm, 3.0 × 50 mm); 8 min chromatography ran 0.037% TFA in water/MeCN (19:1) (solvent A), 0.018% TFA/MeCN (solvent B), and gradient 0–60% (solvent B) over 6.0 min, held at 60% for 1.0 min, and returned to 0% (solvent B) for 1.0 min at a flow rate of 1.0 mL/min; 4 min chromatography ran 0.037% TFA in water/MeCN (19:1) (solvent A), 0.018% TFA/MeCN (solvent B), and gradient 10–80% (solvent B) over 3.0 min, held at 80% for 0.5 min, and returned to 0% (solvent B) for 0.5 min at a flow rate of 1.0 mL/min. All final compounds were >95% pure unless otherwise stated. Nuclear magnetic resonance (NMR) spectra were obtained on Bruker Avance Neo 400 MHz, Bruker Avance Neo 500 MHz, and Bruker Avance 850 MHz instruments (Madison, WI, USA). Chemical shifts are reported in parts per million (ppm, δ), with residual solvent peaks referenced as the internal standard. Coupling constants are reported in Hz. Spin multiplicities are described as s (singlet), br s (broad singlet), d (doublet), t (triplet), q (quartet), p (pentet), and m (multiplet). Data were processed using MestReNova.

*5-chloro-7-(cyclopropylamino)pyrazolo[1,5-a]pyrimidine-3-carbonitrile (***4***).* To a solution of 5,7-dichloropyrazolo[1,5-*a*]pyrimidine-3-carbonitrile (**3**) (5.00 g, 23.47 mmol, 1 *eq*) in EtOH (30 mL) was added cyclopropylamine (12.00 g, 211.24 mmol, 14 mL, 9 *eq*) dropwise at 25 °C. Then, the mixture was stirred at 25 °C for 2 h. The reaction mixture was filtered and the solid was washed with EtOH (4 mL × 2). Compound **4** (5.47 g, 23.03 mmol, 98.1% yield) was obtained as a yellow solid. ^1^H NMR (400 MHz, MeOD-*d*_4_) δ 8.36 (s, 1H), 6.61 (s, 1H), 2.76 (m, 1H), 1.04–0.93 (m, 2H), 0.83–0.74 (m, 2H). LCMS t_R_ = 0.500 min in 1 min chromatography, Chromolith @ Flash RP-18e,25-3mm, MS ESI calcd. for C_10_H_9_ClN_5_^+^ [M+H]^+^ *m*/*z* 234.05, found 234.0.*tert-butyl (S)-(1-(4-amino-5-fluoro-2-nitrophenyl)piperidin-3-yl)(methyl)carbamate* (**7**). To a solution of 2,4-difluoro-5-nitroaniline (**5**) (3 g, 17.23 mmol) in MeCN (25 mL) was added K_2_CO_3_ (7.14 g, 51.69 mmol) and tert-butyl (S)-methyl(piperidin-3-yl)carbamate (**6**) (3.69 g, 17.23 mmol). The mixture was stirred at 100 °C for 12 h. The reaction mixture was concentrated directly. The residue was purified by flash silica gel chromatography (eluent of 0~14% EtOAc in PE) and prep-HPLC (column: Welch Xtimate C18 250 × 50 mm × 10 μm; mobile phase: [water(NH_4_HCO_3_)-ACN]; gradient: 40–85% B over 20 min). Compound **7** (5.1 g, 12.18 mmol, 70.68% yield) was obtained as a red oil. ^1^H NMR (400 MHz, DMSO-*d*_6_) δ 7.28–7.09 (m, 2H), 5.45 (s, 2H), 3.99–3.87 (m, 1H), 3.65–3.57 (m, 2H), 2.96–2.82 (m, 2H), 2.74–2.69 (m, 3H), 1.78–1.73 (m, 2H), 1.59–1.51 (m, 2H), 1.45–1.35 (m, 9H). ^19^F NMR (376 MHz, DMSO-d_6_) δ −124.6. ^1^H-^19^F HOESY spectrum confirmed the regioselectivity of the reaction (Appendix A). LCMS t_R_ = 0.665 min in 1.0 min chromatography, 5-95AB, LCMS ESI calcd. for C_17_H_26_FN_4_O_4_+ [M+H]^+^ 369.19, found 369.1.*tert-butyl (S)-(1-(4-(((benzyloxy)carbonyl)amino)-5-fluoro-2-nitrophenyl)piperidin-3-yl)(methyl)carbamate (***8***).* To a solution of tert-butyl (S)-(1-(4-((3-cyano-7-(cyclopropylamino)pyrazolo[1,5-*a*]pyrimidin-5-yl)amino)-5-fluoro-2-propionamidophenyl)piperidin-3-yl)(methyl)carbamate (**7**) (5.1 g, 13.84 mmol) in THF (50 mL) was added CbzCl (3.54 g, 20.77 mmol, 3 mL) and K_2_CO_3_ (5.7 g, 41.53 mmol) at 0 °C. The mixture was stirred at 25 °C for 12 h. Aqueous NH_4_Cl (20 mL) was added to the reaction mixture. The resulting mixture was extracted with EtOAc (40 mL × 3). The combined organic phase was washed with brine (20 mL) and water (20 mL), dried over anhydrous Na_2_SO_4_, filtered, and concentrated. The residue was purified by flash silica gel chromatography (eluent of 0~19% EtOAc in PE). Compound **8** (7.9 g, 13.80 mmol, 99.69% yield) was obtained as a red oil. ^1^H NMR (400 MHz, DMSO-*d*_6_) δ 9.65 (s, 1H), 8.25–8.14 (m, 1H), 7.39–7.32 (m, 5H), 6.64–6.48 (m, 1H), 5.16 (s, 2H), 3.99 (s, 1H), 3.11–2.99 (m, 2H), 2.93–2.85 (m, 1H), 2.73 (s, 4H), 1.80–1.71 (m, 2H), 1.66–1.58 (m, 2H), 1.40 (s, 9H). LCMS t_R_ = 0.833 min in 1.0 min chromatography, 5-95AB, LCMS ESI calcd. for C_25_H_32_FN_4_O_6_^+^ [M+H]^+^ 503.22, found 503.3.*tert-butyl (S)-(1-(2-amino-4-(((benzyloxy)carbonyl)amino)-5-fluorophenyl)piperidin-3-yl)(methyl)carbamate (***9***)*. To a solution of tert-butyl (S)-(1-(4-(((benzyloxy)carbonyl)amino)-5-fluoro-2-nitrophenyl)piperidin-3-yl)(methyl)carbamate (**8**) (3 g, 5.97 mmol) in EtOH (18 mL) and H_2_O (6 mL) was added Fe (1.00 g, 17.91 mmol) and NH_4_Cl (1.9 g, 35.82 mmol) under a N_2_ atmosphere. The mixture was stirred at 90 °C for 12 h. The reaction mixture was filtered. Aqueous NaHCO_3_ (10 mL) solution was added. The resulting mixture was extracted with EtOAc (35 mL × 3). The combined organic phase was washed with brine (10 mL) and water (10 mL), dried over anhydrous Na_2_SO_4_, filtered, and concentrated. Compound **9** (2.56 g, crude) was obtained as blue oil and used in the next step without further purification.*tert-butyl (S)-(1-(4-(((benzyloxy)carbonyl)amino)-5-fluoro-2-propionamidophenyl)piperidin-3-yl)(methyl)carbamate (***10***).* To a solution of tert-butyl (S)-(1-(2-amino-4-(((benzyloxy)carbonyl)amino)-5-fluorophenyl)piperidin-3-yl)(methyl)carbamate (**9**) (2.5 g, 5.42 mmol) in DCM (25 mL) was added DIPEA (840 mg, 6.50 mmol, 1 mL) and propanoyl chloride (501 mg, 5.42 mmol, 501 μL) at 0 °C under a N_2_ atmosphere. The mixture was stirred at 25 °C for 12 h. The reaction mixture was concentrated. Saturated aqueous NaHCO_3_ (10 mL) solution was added. The resulting mixture was extracted with DCM (20 mL × 3). The mixture was dried over anhydrous Na_2_SO_4_ filtered and concentrated. The residue was purified by flash silica gel chromatography (eluent of 0~19%, EtOAC/PE) to give the product. Compound **10** (1.5 g, 2.34 mmol, 43.12% yield) was obtained as a yellow oil. ^1^H NMR (400 MHz, DMSO-*d*_6_) δ 9.27 (s, 1H), 8.77 (s, 1H), 8.04–7.92 (m, 1H), 7.41–7.32 (m, 5H), 7.05–6.96 (m, 1H), 5.12 (s, 2H), 4.28–4.12 (m, 1H), 2.94–2.81 (m, 2H), 2.71 (s, 3H), 2.62 (d, *J* = 0.8 Hz, 1H), 2.43–2.35 (m, 2H), 1.80–1.67 (m, 3H), 1.66–1.45 (m, 2H), 1.39 (s, 9H), 1.08 (t, *J* = 7.2 Hz, 3H). LCMS t_R_ = 0.703 min in 1.0 min chromatography, 5-95AB, LCMS ESI calcd. for C_28_H_38_FN_4_O_5_^+^ [M+H]^+^ 529.27, found 529.3.*tert-butyl (S)-(1-(4-amino-5-fluoro-2-propionamidophenyl)piperidin-3-yl)(methyl)carbamate (***11***).* To a solution of tert-butyl (S)-(1-(4-(((benzyloxy)carbonyl)amino)-5-fluoro-2-propionamidophenyl)piperidin-3-yl)(methyl)carbamate (**10**) (1.5 g, 2.84 mmol) in MeOH (15 mL) was added Pd/C (700 mg, 657.77 μmol, 10% Pd) under a N_2_ atmosphere. The suspension was degassed and purged with H_2_ (15 Psi) at 25 °C for 12 h. The reaction mixture was filtered. The reaction mixture was concentrated directly. The residue was used in the next step directly. Compound **11** (1 g, 2.41 mmol, 85.05% yield) was obtained as a brown solid. ^1^H NMR (400 MHz, DMSO-*d*_6_) δ 8.62 (s, 1H), 7.54–7.45 (m, 1H), 6.92–6.83 (m, 1H), 4.92 (s, 2H), 4.21–4.03 (m, 1H), 3.62–3.59 (m, 1H), 2.71 (s, 3H), 2.68–2.59 (m, 2H), 2.40–2.33 (m, 2H), 1.79–1.74 (m, 3H), 1.71–1.66 (m, 1H), 1.56–1.49 (m, 1H), 1.38 (s, 9H), 1.12–1.07 (m, 1H), 1.09 (t, *J* = 7.6 Hz, 2H). LCMS t_R_ = 0.596 min in 1.0 min chromatography, 5-95AB, LCMS ESI calcd. for C_20_H_32_FN_4_O_3_^+^ [M+H]^+^ 395.25, found 395.2.*tert-butyl (S)-(1-(4-((3-cyano-7-(cyclopropylamino)pyrazolo[1,5-a]pyrimidin-5-yl)amino)-5-fluoro-2-propionamidophenyl)piperidin-3-yl)(methyl)carbamate (***12***)*. To a solution of tert-butyl (S)-(1-(4-amino-5-fluoro-2-propionamidophenyl)piperidin-3-yl)(methyl)carbamate (**11**) (300 mg, 760.49 μmol) and 5-chloro-7-(cyclopropylamino)pyrazolo[1,5-a]pyrimidine-3-carbonitrile (**4**) (177 mg, 760.49 μmol) in dioxane (5 mL) was added BINAP (71 mg, 114.07 μmol), Pd(OAc)_2_ (25 mg, 114.07 μmol) and Cs_2_CO_3_ (743 mg, 2.28 mmol) at 25 °C. Then, the mixture was degassed and purged with N_2_. The reaction mixture was heated in a microwave at 130 °C for 2 h. The reaction mixture was concentrated directly. The residue was purified by flash silica gel chromatography (eluent of 0~2% MeOH in DCM) and prep-HPLC (column: Welch Ultimate XB-CN 250 × 50 × 10 μm; mobile phase: [Heptane-EtOH(0.1%NH_3_H_2_O)]; gradient: 20%–50% B over 15 min). Compound **12** (900 mg, 1.31 mmol, 77.36% yield) was obtained as a yellow oil. ^1^H NMR (400 MHz, DMSO-d_6_) δ 9.24 (s, 1H), 8.82 (s, 1H), 8.32 (s, 1H), 8.22 (s, 1H), 8.15–8.08 (m, 1H), 7.11–7.05 (m, 1H), 5.95 (s, 1H), 4.28–4.13 (m, 1H), 3.50–3.38 (m, 3H), 2.98–2.86 (m, 2H), 2.70–2.63 (m, 1H), 2.61–2.53 (m, 2H), 2.45–2.37 (m, 2H), 1.83–1.70 (m, 3H), 1.64–1.51 (m, 1H), 1.40 (s, 9H), 1.05 (s, 3H), 0.79–0.74 (m, 2H), 0.72–0.66 (m, 2H). LCMS t_R_ = 0.725 min in 1.0 min chromatography, 5-95AB, LCMS ESI calcd. for C_30_H_39_FN_9_O_3_+ [M+H]^+^ 592.32, found 592.3.*(S)-N-(5-((3-cyano-7-(cyclopropylamino)pyrazolo[1,5-a]pyrimidin-5-yl)amino)-4-fluoro-2-(3-(methylamino)piperidin-1-yl)phenyl)propionamide (***2***)*. To a mixture of tert-butyl (S)-(1-(4-((3-cyano-7-(cyclopropylamino)pyrazolo[1,5-a]pyrimidin-5-yl)amino)-5-fluoro-2-propionamidophenyl)piperidin-3-yl)(methyl)carbamate (**12**) (750 mg, 1.27 mmol) in DCM (5 mL) was added TFA (3.07 g, 26.93 mmol, 2 mL). The mixture was stirred at 25 °C for 2 h. The reaction mixture was concentrated directly. The residue was purified by prep-HPLC (column: Welch SiO_2_ 10u 250 × 300 mm; mobile phase: [Heptane-EtOH(0.1%NH_3_H_2_O)]; gradient:10–25% B over 15 min) and second prep-HPLC (column: Xtimate C18 150 × 40 mm × 10 μm; mobile phase: [water(NH_4_HCO_3_)-ACN]; gradient: 12–52% B over 25 min). Compound **2** (184.9 mg, 375.50 μmol, 30.76% yield, 99.82% purity) was obtained as a white solid. ^1^H NMR (400 MHz, DMSO-d_6_) δ 9.22 (s, 1H), 8.95 (s, 1H), 8.31 (s, 1H), 8.28–8.15 (m, 2H), 7.02 (d, J = 12.0 Hz, 1H), 5.95 (s, 1H), 2.95 (d, J = 10.9 Hz, 1H), 2.89–2.80 (m, 1H), 2.74–2.60 (m, 2H), 2.57 (tt, J = 6.9, 3.6 Hz, 1H), 2.52–2.47 (m, 1H), 2.39 (q, J = 7.5 Hz, 2H), 2.32 (s, 3H), 1.86–1.68 (m, 2H), 1.69–1.55 (m, 1H), 1.43–1.31 (m, 1H), 1.11 (t, J = 7.5 Hz, 3H), 0.82–0.73 (m, 2H), 0.74–0.65 (m, 2H). ^13^C NMR (101 MHz, DMSO-d_6_) δ 172.28, 158.19, 151.39, 151.28, 148.94, 145.65, 128.90, 121.78, 121.65, 120.15, 115.29, 108.19, 107.98, 76.51, 75.87, 57.82, 55.83, 52.25, 34.18, 29.99, 29.35, 23.76, 23.36, 10.27, 6.98. HPLC t_R_ = 3.251 min in 8 min chromatography, purity 99.82%. LCMS t_R_ = 1.465 min in 4 min chromatography, 5-95AB, LCMS ESI calcd. for C_25_H_31_FN_9_O+ [M+H]^+^ 492.26, found 492.2.

### 4.2. NanoBRET Assay

Assays were run using a modified version of the previously published protocols [16,20,24]. HEK293 cells were cultured at 37 °C in 5% CO_2_ in Dulbecco’s modified Eagle medium (DMEM (Gibco, Waltham, MA, USA)) supplemented with 10% fetal bovine serum (VWR/Avantor, Radnor, PA, USA). A transfection complex of DNA (10 μg/mL) was generated consisting of 9 μg/mL carrier DNA (Promega, Madison, WI, USA) and 1 μg/mL CSNK2A-NLuc fusion DNA in Opti-MEM without serum (Gibco). FuGENE HD (Promega) was added at 30 μL/mL to form a lipid–DNA complex. The solution was then mixed and incubated at room temperature for 20 min. The transfection complex was mixed with a 20× volume of HEK293 cells in DMEM/FBS to arrive at a final concentration of 200,000 cells/mL, and 100 μL/well was added to a 96-well plate that was incubated overnight at 37 °C and 5% CO_2_. The following day, the medium was removed via aspiration and replaced with 85 μL of Opti-MEM without phenol red. A total of 5 μL per well of 20× NanoBRET Tracer K10 (Promega) at 10 μM for CSNK2A1 or 5 μM for CSNK2A2 in Tracer Dilution Buffer (Promega N291B) was added to all wells, except the “no tracer” control wells. Test compounds (10 mM in DMSO) were diluted 100× in Opti-MEM media to prepare stock solutions and evaluated at 11 concentrations. A total of 10 μL per well of the 10-fold test compound stock solutions (final assay concentration of 0.1% DMSO) were added. For “no compound” and “no tracer” control wells, DMSO in Opti-MEM was added for a final concentration of 1.1% across all wells; 96-well plates containing cells with NanoBRET Tracer K10 and test compounds (100 μL total volume per well) were equilibrated (37 °C/5% CO_2_) for 2 h. The plates were cooled to room temperature for 15 min. The NanoBRET NanoGlo substrate (Promega) at a ratio of 1:166 to Opti-MEM media in combination with an extracellular NLuc Inhibitor (Promega) diluted at 1:500 (10 μL of 30 mM stock per 5 mL of the Opti-MEM plus substrate) was combined to create a 3× stock solution. A total of 50 μL of the 3× substrate/extracellular NL inhibitor was added to each well. The plates were read within 30 min on a GloMax Discover luminometer (Promega) equipped with a 450 nm BP filter (donor) and 600 nm LP filter (acceptor) using 0.3 s of integration time. Raw milliBRET (mBRET) values were obtained by dividing the acceptor emission values (600 nm) by the donor emission values (450 nm) and multiplying by 1000. Averaged control values were used to represent complete inhibition (no tracer control: Opti-MEM + DMSO only) and no inhibition (tracer only control: no compound, Opti-MEM + DMSO + Tracer K10 only) and were plotted alongside the raw mBRET values. The data were first normalized and then fit using the Sigmoidal 4PL binding curve in Prism Software version 10.2.0 to determine IC_50_ values.

### 4.3. In-Cell Selectivity Profiling Using NanoBRET K192 Assay

The K192 selectivity assay was run according to the Draft Promega technical manual, NanoBRET^TM^ Target Engagement K192 Kinase Selectivity System. Reagents were supplied by Promega (Promega NP 4101). For the assay, DNA from the prepared kinase vector panel plates A and B were mixed with Fugene in 96-well plates (Corning 3917) and incubated at room temperature for 30 min. The NanoLuc^®^ Low control vector pNL1.1.CMV [Nluc/CMV] Vector (Cat.# N1091) and the transfection control vector NanoLuc^®^-HIPK2 Fusion Vector (Cat.# NV3221) were used as controls.

HEK293 cells were grown to 75–95% confluency in DMEM (Gibco 11995-065) supplemented with FBS (Avantor 97068-085) at 37 °C in 5% RH. On the first day of the assay, cells were harvested and resuspended in Opti-MEM (Gibco 11058-021) supplemented with 1% FBS (Avantor 97068-085) at 2.5 × 10^5^ cells per mL. A volume of 60 μL of cell suspension was mixed with 10 μL of prepared DNA (10X concentration) and 30 μL of Fugene (30 μL/mL in Opti-MEM) as outlined by Promega and incubated overnight at 37 °C in a 5% CO_2_ incubator.

On day two of the assay, 5 μL of 20X K10 tracer was prepared and added at concentrations recommended by Promega. Then, 10 μL of test compounds at 100 μM in Opti-MEM (diluted from a 10 mM solution in DMSO) was added to the test wells while an equivalent volume of Opti-MEM was added to the high-control wells. Plates were kept at 37 °C in a 5% CO_2_ incubator for two hours. After two hours, plates were allowed to equilibrate to room temperature for 15 min. A solution of 3X Complete Substrate plus Inhibitor Solution was freshly prepared, consisting of a 1:166 dilution of NanoBRET™ Nano-Glo^®^ Substrate plus a 1:500 dilution of Extracellular NanoLuc^®^ Inhibitor in Opti-MEM^®^ medium without serum or phenol red. A 50 μL volume of the 3X Complete Substrate plus Inhibitor Solution was added to each assay well, including control wells. After 2–3 min, the plate was shaken at 300 RPM for 10 s, and the donor emission wavelength (450 nm) and acceptor emission wavelength (610 nm) were measured using the Glomax^®^ Discover System.

As a quality check, the donor signal-to-background ratio was calculated for each individual kinase by dividing the mean donor signal for each kinase by the mean donor signal for the signal-to-background control wells. Fractional occupancy of the test drug for each kinase was determined using the following formula:Occupancy (%) = [1 − (Sample − Bottom)/(Top − Bottom)] × 100
where

Sample = Mean BRET value across all Sample (tracer + compound) wells for an individual kinase;

Top = Mean BRET value across all Top (tracer + vehicle) control wells for an individual kinase;

Bottom = Mean BRET value of NanoLuc^®^ control wells (calculated either on a plate-by-plate basis or across the entire experiment).

### 4.4. MHV Assay

DBT cells were cultured at 37 °C in Dulbecco’s modified Eagle medium (DMEM; Sigma-Aldrich, St. Louis, MO, USA) supplemented with 10% fetal bovine serum (FBS; Avantor, Randor, PA, USA) and penicillin and streptomycin (Sigma-Aldrich, St. Louis, MO, USA). DBT cells were plated in 96-well plates to be 80% confluent at the start of the assay. Test compounds or positive control EIDD-1931 were diluted to 15 μM in DMEM. Serial 4-fold dilutions were made in DMEM, providing a concentration range of 15–0.22 μM. Media were aspirated from the DBT cells, and 100 μL of the diluted test compounds was added to the cells for 1 h at 37 °C. After 1 h, MHV-nLuc5 was added at an MOI of 0.1 in 50 μL of DMEM so that the final concentration of the first dilution of the compound was 10 μM (T = 0). After 10 h, the media were aspirated, and the cells were washed with PBS and lysed with passive lysis 5X buffer (Promega, Madison, WI, USA) for 20 min at room temperature. Relative light units (RLUs) were measured using a luminometer (Promega; GloMax, Madison, WI, USA). Triplicate data were analyzed in GraphPad Prism version 9.4.1 for macOS to generate IC_50_ values. A dose–response of EIDD-1931 was used as a positive control for the assay; each plate also contained a set of wells treated with EIDD-1931 at the IC_50_ for the assay (1.2 µM).

### 4.5. SARS-CoV-2 Assay

Human lung epithelial A549-ACE2 cells were cultured in DMEM (Gibco, Thermo Fisher Scientific, Waltham, MA, USA) containing 10% heat-inactivated FBS (Biotechne, R&D Systems, Minneapolis, MN, USA), nonessential amino acids (Gibco, Thermo Fisher Scientific, Waltham, MA, USA), and pen strep (Gibco, Thermo Fisher Scientific, Waltham, MA, USA). A549-ACE2 cells were seeded at 20,000 cells per well in a 96-well solid black plate 1 day prior to infection. To assay the drug effect, cells were pretreated with the drug for 1 h and then infected with SARS-CoV-2, with the drug maintained during the infection. Then, 2 h after infection, the supernatant was removed, monolayers were rinsed with PBS, and medium containing the drug was added to each well. At 48 h post-infection start, Nano-glo was added to each well as per the manufacturer’s protocol (Promega, Madison, WI, USA) and RLUs were measured using a Promega GloMax.

### 4.6. Kinetic Solubility

Phosphate buffered saline (50 mL, PBS, Fisher (Pittsburgh, PA, USA), pH 7.4) was added to HPLC-grade H_2_O (450 mL) for a total dilution factor of 1:10 and a final PBS concentration of 1×. The test compound (6 μL) as a 10 mM DMSO stock solution was combined with the aqueous PBS solution (294 μL) for 50-fold dilution in a Millipore solubility filter plate with a 0.45 μM polycarbonate filter membrane using a Hamilton Starlet liquid handler. The final DMSO concentration was 2.0%, and the maximum theoretical compound concentration was 200 μM. The filter plate was heat-sealed for a 24 h incubation. The sample was placed on a rotary shaker (200 rpm) for 24 h at ambient temperature (21.6–22.8 °C) and then vacuum-filtered. All filtrates were injected into a chemiluminescent nitrogen detector for quantification. The equimolar nitrogen response of the detector was calibrated using standards that span the dynamic range of the instrument from 0.08 to 4500 μg/mL nitrogen. The filtrates were quantified with respect to this calibration curve. The calculated solubility values were corrected for background nitrogen present in DMSO and the media used to prepare the samples.

### 4.7. CellTiter-Glo Cytotoxicity Assay

A549-ACE2 cells were maintained in low-glucose DMEM (Gibco, Waltham, MA, USA) supplemented with 10% FBS, 1% NEAA, and 1% L-glutamine. No antibiotics were used. Cells were plated at 2000 cells/well in a 384-well plate (Costar, Glendale, AZ, USA) and incubated overnight (37 °C, 5% CO_2_) before adding the compound. Compounds were added in quadruplicate and incubated for 48 h. DMSO percentage was constant across all concentrations of the compound. Cell viability was measured using CellTiter-Glo2 (Promega, Madison, WI, USA) and the luminescence signal was read on a GloMax plate reader (Promega, Madison, WI, USA). Dose–response analysis was performed using GraphPad Prism.

### 4.8. Crystallography

CSNK2A1 expression and purification were performed as described previously [24,38,39]. Briefly, transformed BL21(DE3) cells were grown in Terrific Broth medium containing 50 mg/mL kanamycin. Protein expression was induced at OD_600_ using 0.5 mM isopropyl-thio-galactopyranoside (IPTG) at 18 °C for 12 h. Cells expressing His6-tagged CSNK2A1 were lysed in lysis buffer containing 50 mM HEPES, pH 7.5, 500 mM NaCl, 25 mM imidazole, 5% glycerol, and 0.5 mM Tris(2 carboxyethyl)phosphine (TCEP) by sonication. After centrifugation, the supernatant was loaded onto a Nickel–Sepharose column equilibrated with 30 mL lysis buffer. The column was washed with 60 mL lysis buffer. Proteins were eluted using an imidazole step gradient (50, 100, 200, 300 mM). Fractions containing protein were pooled together and dialyzed overnight using 1L of final buffer (25 mM HEPES, pH 7.5, 500 mM NaCl, 0.5 mM TCEP) at 4 °C. Additionally, TEV protease was added (protein/TEV 1:20 molar ratio) to remove the tag. The next day, the protein solution was loaded onto Nickel–Sepharose column beads again to remove the TEV protease and cleaved Tag. The combined flow-through fraction and the wash fraction (25 mM imidazole) containing the protein were concentrated to approximately 4–5 mL and loaded onto the Superdex 75 16/60 Hi-Load gel filtration column equilibrated with the final buffer. The protein was concentrated to approximately 9 mg/mL.

CSNK2A1 was crystallized using the sitting drop vapor diffusion method by mixing protein (9 mg/mL) and well solutions in 2:1, 1:1, and 1:2 ratios. The reservoir solution contained 0.2 M ammonia sulphate, 0.1 M bis-tris pH 5.5, and between 23 and 26% (*v*/*v*) PEG 3350. Complex structures were achieved by soaking grown apo crystals for at least 24 h with the desired inhibitor dissolved in reservoir solution. The final concentration of the inhibitor was approximately 0.5 mM.

Diffraction data were collected at beamline X06SA (Paul Scherrer Institut, Villigen, Switzerland) at a wavelength of 1.0 Å at 100 K. The reservoir solution supplemented with 20% ethylene glycol was used as a cryoprotectant. Data were processed using XDS [40] and scaled with aimless [41]. The PDB structure with the accession code 6Z83 [24] was used as an initial search MR model using the program MOLREP [42]. The final model was built manually using Coot [43] and refined with REFMAC5 [44]. Crystallographic refinement statistics are provided in Appendix A.

### 4.9. Liver Microsomal Stability Assay

Compounds **1** (0 mM DMSO stock solutions) were diluted to 2.5 mM with DMSO and again to 0.5 mM with MeCN to give a final solution containing a 0.5 mM compound in 1:4 DMSO/MeCN. Liver microsomes from male CD-1 mice were sourced from Xenotech (Kansas City, KS, USA). A reaction plate was prepared by adding 691.25 μL and prewarmed (37 °C) microsomal solution (0.63 mg/mL protein and 1.3 mM EDTA in potassium phosphate buffer made by mixing ~250 mL of 100 mM K_2_HPO_4_ with ~65mL of KH_2_PO_4_ until the buffer reaches a pH of 7.4) to an empty well of a 96-well plate and maintained at 37 °C. The diluted 0.5 mM compound (8.75 μL) was added to the microsomal solution in the reaction plate and mixed thoroughly by repeated pipetting to give a final assay concentration of 5.0 μM. The resulting solutions were preincubated for 5 min at 37 °C and then dispensed into T = 0 and incubation plates. For the T = 0 plates, an aliquot (160 μL) of each reaction solution was added to an empty well of a 96-well plate as an exact replicate of the reaction plate. Cold (4 °C) MeOH (400 μL) was added to each well and mixed thoroughly by repeated pipetting. NADPH regeneration solution (40 μL) was added to each well and mixed thoroughly by repeated pipetting. For the T = 30 min incubation plate, NADPH (95 μL) was added to the remaining solution (microsomes + test compound) in each well in the previously prepared reaction plate to initiate the reaction. The plate was sealed and incubated at 37 °C for 30 min. An aliquot (100 μL) was removed from each well at the desired time point and dispensed into a well of a 96-well plate. Cold (4 °C, 200 μL) MeOH was added to quench the reaction. All plates were sealed, vortexed, and centrifuged at 3000 rpm and 4 °C for 15 min, and the supernatants were transferred for analysis by LC-TOFMS. The supernatant (20 μL) was injected into an AQUASIL C18 column and eluted using a fast-generic gradient program. TOFMS data were acquired using Agilent 6538 Ultra High Accuracy TOF MS in extended dynamic range (*m*/*z* 100–1000) using generic MS conditions in positive mode. Following data acquisition, exact mass extraction and peak integration were performed using MassHunter Software version 12.1 (Agilent Technologies, Santa Clara, CA, USA). The stability of the compound was calculated as the percentage of the unchanged parent remaining at T = 30 min relative to the peak area at T = 0 min.

To determine CL_int_, aliquots of 50 μL were taken from the reaction solution at 0, 15, 30, 45, and 60 min. The reaction was stopped by the addition of four volumes of cold MeCN with IS (100 nM alprazolam, 200 nM imipramine, 200 nM labetalol, and 2 μM ketoprofen). Samples were centrifuged at 3220× *g* for 40 min, and 90 μL of the supernatant was mixed with 90 μL of ultrapure H_2_O and then used for LC-MS/MS analysis. Peak areas were determined from extracted ion chromatograms and the slope value, k, was determined by linear regression of the natural logarithm of the remaining percentage of the parent drug vs. incubation time curve. The intrinsic clearance (CL_int_ in μL/min/mg) was calculated using the relationship CL_int_ = kV/N where V is the incubation volume and N is the amount of protein per well.

### 4.10. Hepatocyte Stability Assay

Human cryopreserved hepatocytes were supplied by BioIVT (lot QZW, 10 pooled donors, Westbury, NY, USA). Mouse cryopreserved hepatocytes were supplied by BioIVT (lot ZPG, pooled male CD-1). Vials of cryopreserved hepatocytes were removed from storage and thawed in a 37 °C water bath with gentle shaking, and then the contents were poured into a 50 mL thawing medium conical tube. Vials were centrifuged at 100× *g* for 10 min at room temperature. The thawing medium was aspirated, and hepatocytes were resuspended with a serum-free incubation medium to yield ~1.5 × 10^6^ cells/mL. Cell viability and density were counted using AO/PI fluorescence staining, and then cells were diluted with a serum-free incubation medium to a working cell density of 0.5 × 10^6^ viable cells/mL. Aliquots of 198 μL of hepatocytes were dispensed into each well of a 96-well noncoated plate. The plate was placed in an incubator for approximately 10 min. Aliquots of 2 μL of the 100 μM test compound in duplicate and positive control were added into the respective wells of the noncoated 96-well plate to start the reaction. The final concentration of the test compound was 1 μM. The plate was placed in an incubator for the designed time points. Contents (25 μL) were transferred and mixed with six volumes (150 μL) of cold MeCN with IS (100 nM alprazolam, 200 nM labetalol, 200 nM caffeine, and 200 nM diclofenac) to terminate the reaction at time points of 0, 15, 30, 60, 90, and 120 min. Samples were centrifuged for 45 min at 3220× *g*, an aliquot of 100 μL of the supernatant was diluted with 100 μL of ultrapure H_2_O, and the mixture was used for LC-MS/MS analysis. Peak areas were determined from extracted ion chromatograms, and the slope value, k, was determined by linear regression of the natural logarithm of the remaining percentage of the parent drug vs. incubation time curve. The intrinsic clearance (CL_int_ in μL/min/10^6^ cells) was calculated using the relationship CL_int_ = kV/N, where V is the incubation volume (0.2 mL) and N is the number of hepatocytes per well (0.1 × 10^6^ cells). Scaling factors to convert CL_int_ from μL/min/10^6^ cells to mL/min/kg were 2540 (human hepatocytes) and 11,800 (mouse hepatocytes).

### 4.11. In Vitro Inhibition of Phosphorylation of EIF2S2

A549ACE2 cells were seeded at 150,000 cells per well in a 6-well plate in DMEM high-glucose containing 10% fetal bovine serum, 1X non-essential amino acids, and 2 mM L-glutamine, and allowed to adhere overnight. Cells were treated with compound **2** (1 μM or 5 μM) or 0.05% DMSO (vehicle control) for 24 h. Cells were lysed with RIPA buffer and sonicated at 40% for 10 s on ice. Lysates were quantified using Pierce Rapid Gold BCA Protein Assay kit and 20 mg protein was run on 4–20% Tris-Glycine gels and transferred to PVDF. Blots were blocked in 5% Milk in 1X TBST (0.1% Tween 20) for 1 h at room temperature, washed with 1X TBST, and incubated with primary antibodies in 5% BSA in 1X TBST for 10 h at room temperature; 1:10,000 phospho-EIF2S2 P-S2 (from Laszlo Gyenis from the David Litchfield group), 1:200 EIF2S2 (Novus Biologicals, H00008894-M09, Centennial, CO, USA), 1:5000 GAPDH (Proteintech, 10494-1-AP, Rosemont, IL, USA). Bands were quantified using Fiji and normalized to GAPDH. Phosphorylated protein was then normalized to its corresponding total protein. Plots were generated using GraphPad Prism version 9.4.1 for macOS (N = 2).

### 4.12. Pharmacokinetic Studies

In the 24 h PK study, male CD-1 mice (6–8 weeks old, 20–30 g) were dosed by intravenous (i.v.), oral (p.o.), or intraperitoneal (i.p.) administration of compound **2**. For i.v. administration, a single dose of compound **2** (3 mg/kg) was administered as 5 mL/kg of a 0.6 mg/mL solution in DMSO/PEG-400/Water (*v*/*v*/*v*, 10:30:60) to six mice. For p.o. administration, a single dose of compound **2** (10 mg/kg) was administered as 10 mL/kg of a 1 mg/mL solution in DMSO/PEG-400/Water (*v*/*v*/*v*, 10:30:60) to six mice. For i.p. administration, a single dose of compound **2** (10 mg/kg) was administered as 10 mL/kg of a 1 mg/mL solution in DMSO/PEG-400/Water (*v*/*v*/*v*, 10:30:60) to six mice. The mice had free access to water and food. At 0.083, 0.25, 0.5, 1, 2, 4, 8, 12, and 24 h post dose administration, 0.03 mL of blood was collected from the dorsal metatarsal vein of three mice at each time point. Each blood sample was transferred into a plastic microcentrifuge tube containing EDTA-K_2_ anticoagulant and mixed well with the anticoagulant then placed on ice prior to centrifugation at 4000× *g* for 5 min at 4 °C to obtain plasma. The samples were stored in a freezer at −75 ± 15 °C prior to analysis. At 12 h post dose administration and 24 h post dose administration, three mice were anesthetized using increasing concentrations of CO_2_, and lung samples were collected. Lung samples were quickly frozen in an ice box and stored at −75 ± 15 °C. Prior to analysis, all lung samples were weighed and homogenized with phosphate-buffered saline (PBS) in a lung weight (g)/buffer volume (mL) ratio of 1:3. The final concentration of the compound was calculated by multiplying the detected concentration by a dilution factor of four.

In the 5 h PK study investigating the effect of co-dosing EA and 1-ABT, two cohorts of male CD-1 mice (6–8 weeks old, 20–30 g) were dosed via intraperitoneal (i.p.) administration of compound **2** (10 mg/kg) as a 10 mL/kg volume of a 1.0 mg/mL solution in DMSO/PEG-400/saline (*v*/*v*/*v*, 10:30:60). EA (30 mg/kg) was administered i.p. as a 1.0 mL/kg volume of a 30 mg/mL solution in NMP/PEG-400/water (*v*/*v*/*v*, 10:60:30) as a 6 h pretreatment and again at the time of dosing with compound **2**. 1-ABT (100 mg/kg) was administered p.o. as a 10 mL/kg volume of a 10 mg/mL solution in saline as a 2 h pretreatment. No co-treatment with EA or 1-ABT was applied to the first cohort. Co-treatment with EA (30 mg/kg) and 1-ABT (100 mg/kg) was applied in the second cohort. Two mice were dosed in each cohort. The mice had free access to water and food. Blood (0.03 mL) was collected from the dorsal metatarsal vein at 0.5, 1, 3, and 5 h time points. Each blood sample was transferred into a plastic microcentrifuge tube containing EDTA-K_2_ anticoagulant. Blood samples were centrifuged at 4000× *g* for 5 min at 4 °C to obtain plasma. The samples were stored in a freezer at −75 ± 15 °C prior to analysis. Plasma samples from the two mice from each cohort and each time point were pooled together for analysis. At 5 h post dose administration, the mice were anesthetized using increasing concentrations of CO_2_, and lung samples were collected. Lung samples were quickly frozen in an ice box and stored at −75 ± 15 °C. Prior to analysis, all lung samples were weighed and homogenized with phosphate-buffered saline (PBS) in a lung weight (g)/buffer volume (mL) ratio of 1:3. The final compound concentration was calculated by multiplying the detected concentration by a dilution factor of four.

In the multi-dose PK study, three male Balb/c mice (6–8 weeks old, 20–30 g) were dosed via oral (p.o.) administration of four doses of compound **2** (30 mg/kg) as a 10 mL/kg volume of 3 mg/mL solution in DMSO/PEG-400/Water (*v*/*v*/*v*, 10:30:60) every 12 h. At 2, 6, 12, 14, 24, 26, 36, 38, and 48 h after the first dose, 0.025 mL of blood was collected from the dorsal metatarsal vein at each time point. Each blood sample was transferred into a plastic microcentrifuge tube containing EDTA-K_2_ anticoagulant and mixed well with the anticoagulant, then placed on ice prior to centrifugation at 4000× *g* for 5 min at 4 °C to obtain plasma. The samples were stored in a freezer at −75 ± 15 °C prior to analysis. At 48 h post dose administration, mice were anesthetized using increasing concentrations of CO_2_, and lung samples were collected. Lung samples were quickly frozen in an ice box and stored at −75 ± 15 °C. Prior to analysis, all lung samples were weighed and homogenized with phosphate-buffered saline (PBS) in a lung weight (g)/buffer volume (mL) ratio of 1:3. The final compound concentration was calculated by multiplying the detected concentration by a dilution factor of four.

Concentrations of the test compound in the plasma samples were determined using a Prominence LC-MS/MS system with a DGU-20A5R(C) degasser, LC-30AD pump, CBM-20A system controller, SIL-20AC HT autosampler; Rack changer II, HALO 90A ES-CN (2.7 μm, 2.1 × 50 mm) column, and AB Sciex Triple Quan 5500 LC/MS/MS instrument. The mobile phase was 5–95% MeCN in H_2_O, with 0.1% formic acid. The desired serial concentrations of working solutions were achieved by diluting the stock solution of the analyte with 50% acetonitrile in water. A series of 10 µL working solutions (0.5, 1, 2, 5, 10, 50, 100, 500, 1000 ng/mL) were added to 10 μL of blank plasma or lung homogenate to achieve calibration standards of 0.5, 1, 2, 5, 10, 50, 100, 500, 1000 ng/mL in a total volume of 20 μL. A total of 5 quality control samples at 1 ng/mL, 2 ng/mL, 5 ng/mL, 100 ng/mL, and 800 ng/mL were prepared independently of those used for the calibration curves in the same manner. A 20 μL standard, 20 μL QC sample, and 20 μL of an unknown sample (10 µL plasma or lung homogenate with 10 µL blank solution) were added to 200 μL of acetonitrile containing internal standard mixture for precipitating protein. Then, the samples were vortexed for 30 s. After centrifugation at 4 °C and 4000 rpm for 15 min, the supernatant was diluted with ultrapure water at a ratio of 1:2 (*v*/*v*, 1:2), then 20 µL of diluted supernatant was injected into the LC/MS/MS system for quantitative analysis. PK parameters were calculated from the mean plasma concentration versus time using a noncompartmental model in WinNonlin Phoenix version 8.3.

### 4.13. In Vivo SARS-CoV-2 Inhibition

All mouse studies were conducted using protocols approved by the Institutional Animal Care and Use Committee (IACUC) of the University of North Carolina at Chapel Hill (Protocols 20-128 approved May 2020 and 23-103 approved May 2023).

Balb/c mice (female, 8–10 weeks old, from Envigo, Inotiv, Indianapolis, IN, USA) were treated with compound **2** at 30 mg/kg p.o. in a vehicle (DMSO/PEG-400/Water (*v*/*v*/*v*), 10:30:60; Sigma, St. Louis, MO, USA). Mice were dosed every 12 h for a total of 36 h (3 doses). Twelve hours after the first dose, mice were anesthetized with 50 mg/kg ketamine + 5 mg/kg xylazine and intranasally infected with 1 × 10^4^ PFU of mouse-adapted coronavirus, SARS-CoV-2 MA10 [45,46], in a 50 μL volume that was pipetted into the nares of each mouse. Post-challenge, mice were monitored, weighed, and scored for clinical signs; they were euthanized 24 h post-infection using an overdose of isoflurane anesthesia (Baxter, Deerfield, IL, USA). Blood was collected via cardiocentesis, and lung lobes were collected for downstream analysis.

Infectious viral loads were measured using the plaque assay. One day prior to the assay, Vero cells were seeded at 2 × 10^5^ cells per well in 12-well plates. Titers were measured from superior and middle lung lobes that were homogenized in 0.5 mL of media (DMEM (Gibco, Thermo Fisher Scientific, Waltham, MA, USA) + 5% FBS (Biotechne, R&D Systems, Minneapolis, MN, USA) + 1% L-glutamine (Gibco, Thermo Fisher Scientific, Waltham, MA, USA)) at 6000 rpm for 40 s using a Roche MagNA Lyser homogenizer (Roche, Indianapolis, IN, USA). Cell debris was removed by centrifugation for 1 min at full speed. A 50 μL volume of the supernatant of the clarified homogenate was added to 450 μL of dilution media (DMEM + 5% FBS + 1% L-glutamine media). Homogenates were used to create tenfold serial dilutions (10^−1^ to 10^−6^). Approximately 200 μL of each dilution was pipetted onto the previously plated Vero cells and incubated at 37 °C. To ensure even distribution across each well, the plates were rocked every 15 min. After 1 h, 2 mL of overlay (50:50 mixture of 2.5% carboxymethylcellulose (Sigma, St. Louis, MO, USA) and 2X alpha MEM (Gibco, Thermo Fisher Scientific, Waltham, MA, USA) containing 6% FBS (Biotechne, R&D Systems, Minneapolis, MN, USA) + 2% penicillin/streptomycin (Gibco, Thermo Fisher Scientific, Waltham, MA, USA) + 2% L-glutamine (Gibco, Thermo Fisher Scientific, Waltham, MA, USA) + 2% HEPES (Gibco, Thermo Fisher Scientific, Waltham, MA, USA)) was added to each well. After incubation for 4 days at 37 °C and 5% CO_2_, an equal volume of 4% paraformaldehyde was added to each well and the cells were allowed to fix overnight. The fixative was removed, wells were rinsed with water to remove residual overlay, and 0.25% crystal violet was added to each well. Visible plaques were counted and averaged between two technical replicate wells and used to calculate plaque-forming units (pfus) per lung tissue. The limit of detection (LOD) for the assay was determined to be 12.5 pfus/lung tissue, and samples that yielded no plaques were assigned a value of 6.25, half of the LOD.

### 4.14. In Vivo CSNK2 Inhibition

All mouse studies were conducted using protocols approved by the Institutional Animal Care and Use Committee (IACUC) of the University of North Carolina at Chapel Hill (Protocols 20-128 approved May 2020 and 23-103 approved May 2023).

Balb/c mice (female, 8–10 weeks old, from Envigo, Inotiv, Indianapolis, IN, USA) were treated with compound **2** at 30 mg/kg p.o. in a vehicle (DMSO/PEG-400/Water (*v*/*v*/*v)*, 10:30:60). Mice were dosed every 12 h for a total of 36 h (3 doses). At the determined time points, mice were euthanized using an overdose of isoflurane (Baxter, Deerfield, IL, USA). Blood was collected via cardiac puncture and lung lobes were collected for downstream processing and analysis. Recovered blood was placed in EDTA tubes, spun at 5000× *g* in a microcentrifuge and the recovered plasma transferred to a clean tube and frozen at −80 °C until analysis. The left lung lobe was added to a 2 mL O-ring skirted tube containing 750 μL of 1 × PBS containing Phos Stop (per the manufacturer’s recommendation, Roche) and glass beads. Lungs were homogenized for 60 s at 6000 rpm in a Roche MagNA Lyser. Homogenates were centrifuged for 5 min at 10,000 rpm and 500 μL of the homogenate was transferred to a clean tube and frozen at −80 °C until analysis.

A Halt Protease Inhibitor cocktail (ThermoFisher, 78429, Waltham, MA, USA) was added to mouse lung homogenates prior to sonication at 40% for 10 s on ice. Samples were quantified using the Piece Rapid Gold BCA Protein Assay kit (ThermoFisher, A53226, Waltham, MA, USA), and 50 μg protein was run on 4–20% Tris–Glycine gels and transferred to PVDF. Blots were blocked in 5% Milk in 1X TBST (0.1% Tween 20) for 1 h at room temperature, washed with 1X TBST, and incubated with primary antibodies in 5% BSA in 1X TBST for 10 h at room temperature; 1:10,000 phospho-EIF2S2 P-S2 (provided by Laszlo Gyenis from the David Litchfield group), 1:200 EIF2S2 (Novus Biologicals, H00008894-M09, Centennial, CO, USA), 1:1000 phospho-AKT Ser129 (Cell Signaling Technology, 13461, Danvers, MA, USA), 1:1000 AKT (Cell Signaling Technology, 2920), 1:1000 GAPDH (Proteintech, 10494-1-AP), 1:1000 Transferrin (Proteintech, 17435-1-AP). Bands were quantified using Fiji [47] and normalized to GAPDH. Phosphorylated protein was then normalized to its corresponding total protein. GraphPad Prism version 9.4.1 for macOS was used to plot averages (N = 3 for each treatment) and assess statistical significance using an unpaired *t*-test with Welch’s correction.

## Data Availability

The co-crystal structure of CSNK2A1 with compound **2** (PDB ID: 9FYF) has been deposited in the PDB. Data are contained within the article or Appendix A. The original contributions presented in the study are included in the article/Appendix A, further inquiries can be directed to the corresponding author/s.

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
