# Peer review of "Strategic Fluorination to Achieve a Potent, Selective, Metabolically Stable, and Orally Bioavailable Inhibitor of CSNK2"

_molecules, 2024, doi:10.3390/molecules29174158_

Round 1
Reviewer 1 Report
Comments and Suggestions for Authors
In this manuscript, Willson and co-workers report a strategy to improve the potency, selectivity, metabolic stability and oral bioavailability of inhibitors of casein kinase 2 (CSNK2) by fluorination of a pyrazolo[1,5-a]pyrimidine compound. By strategically inserting a fluorine atom on an electron-rich phenyl ring of the pyrazolo-pyrimidine scaffold, the authors synthesized the compound 2, which was demonstrated to possess improved metabolic stability while maintaining an excellent potency against CSNK2, and submicromolar antiviral potency in vitro with favourable solubility and remarkable selectivity for CSNK2. The high binding affinity of compound 2 to CSNK2 was further confirmed by co-crystal structure characterization. Although the compound 2 did not show antiviral activity in vivo, this manuscript provides a rational strategy for improving the metabolic stability and orally bioavailability of CSNK2 inhibitors by chemical modification of the pyrazolo-pyrimidine scaffold. Thus, I would like to recommend publication of this manuscript in Molecules subjected to a major revision.
Specific comments:
1. My most concern on this work is about the authors’ interpretation on the absence of in vivo antiviral activity of the newly synthesized compound 2. I understand that the in vivo antiviral activity of a potential antiviral drug depends on many elements, such as metabolic stability, drug transportation and organ-targeting, and microenvironment of infected organs or tissues, and so on. Authors explained that the absence of in vivo antiviral activity of compound 2 was probably attributed to its lack of prolonged CSNK2 inhibition. They also mentioned that the previous studies have suggested that the in vivo free drug concentration may need to remain above the EC90 levels of the drug determined in cellular assays for therapeutic effect. However, firstly, the authors should explain why compound 2 lacks prolonged CSNK2 inhibition in vivo. This is due to in vivo metabolism of the compound, or due to competition of high concentration of ATP in vivo? Secondly, if the authors believed that the in vivo concentration of compound 2 was insufficient for its in vivo antiviral activity, why did not they increase the dose of the compound? This is because the solubility of the compound is not good enough? With regard to this, I suggest that the authors explain further the absence of in vivo antiviral activity of compound 2 by more experiments either to prove why the compound 2 can maintain transient inhibitory activity against CSNK2, or to increase the administrative dose of compound 2 so as to achieve a significant in vivo antiviral activity as they expected.
2. Other minor comments:
- In table 1: why the authors did only a singlicate assay for determination of IC50 values of compound 2 towards CSNK2A1/A2? The same question is also for the NanoBRET K192 assay.
- Line 342: “sdosing” should be “dosing”.
Author Response
- My most concern on this work is about the authors’ interpretation on the absence of in vivo antiviral activity of the newly synthesized compound 2. I understand that the in vivo antiviral activity of a potential antiviral drug depends on many elements, such as metabolic stability, drug transportation and organ-targeting, and microenvironment of infected organs or tissues, and so on. Authors explained that the absence of in vivo antiviral activity of compound 2 was probably attributed to its lack of prolonged CSNK2 inhibition. They also mentioned that the previous studies have suggested that the in vivo free drug concentration may need to remain above the EC90 levels of the drug determined in cellular assays for therapeutic effect. However, firstly, the authors should explain why compound 2 lacks prolonged CSNK2 inhibition in vivo. This is due to in vivo metabolism of the compound, or due to competition of high concentration of ATP in vivo? Secondly, if the authors believed that the in vivo concentration of compound 2 was insufficient for its in vivo antiviral activity, why did not they increase the dose of the compound? This is because the solubility of the compound is not good enough? With regard to this, I suggest that the authors explain further the absence of in vivo antiviral activity of compound 2 by more experiments either to prove why the compound 2 can maintain transient inhibitory activity against CSNK2, or to increase the administrative dose of compound 2 so as to achieve a significant in vivo antiviral activity as they expected.
- Compound 2 lacks sustained CSNK2 inhibition in vivo due to insufficient plasma concentrations. In our discussion and in Figure 5, we have added a comparison of the free drug concentration of compound 2 with the CSNK2 IC90 and IC50 determined in the NanoBRET assay. The free plasma concentration of 2 was above the CSNK2 NanoBRET IC50 for the entirety of the 48 h dosing period. However, since the NanoBRET IC90 was only reached at the Cpeak concentrations (2, 14, 26, and 38 h timepoints), CSNK2 inhibition would only be achieved transiently during the in vivo antiviral study. We believe this explains the transient CSNK2 inhibition.
- CYP450 and GST-mediated metabolism was a major contributor to metabolism of this series. While we have mitigated CYP450-mediated metabolism in vivo, it is possible that there were other modes of clearance for the compound which were still present.
- The high cellular concentration of ATP was not a major factor because our CSNK2 pIC50 determinations were performed in live cell NanoBRET assays and SARS-CoV-2 antiviral activity was measured in a phenotypic assay. Both of these assays are performed in cells that maintain physiologically relevant concentrations of ATP.
- During our studies of CSNK2 inhibitors, we have found that high doses of related pyrazolo[1,5‑a]pyrimidine inhibitors induce clinical symptoms in mice, and we are aware of the risks involved of increasing the dose. For compound 2, a 10 mg/kg dose at using i.p. or p.o. routes of administration were well tolerated in mice with no adverse symptoms, while a 30 mg/kg dose p.o. dose b.i.d. led to slight (mean of 11%) weight loss in mice with no other symptoms. To avoid possible toxicity effects and for ethical reasons, we decided not to pursue higher doses of compound 2 given these risks.
- Other minor comments:
- In table 1: why the authors did only a singlicate assay for determination of IC50 values of compound 2 towards CSNK2A1/A2? The same question is also for the NanoBRET K192 assay.
- We believe that by performing a dose-response determination for both CSNK2A1 and CSNK2A2, even though they are in singlicate, should provide sufficient evidence that compounds 1 and 2 are potent CSNK2 inhibitors. In addition, the dose-response data collected on both CSNK2A1 and CSNK2A2 did not show any major difference. This is consistent with the results we have obtained for other analogues of the pyrazolo[1,5‑a]pyrimidine series in our previous publications (Ref. 16, 20, 24, 25).
- The NanoBRET K192 assay is a resource-intensive assay best used for wide kinome profiling at a single concentration. We understand the limitations of a singlicate determination, so any kinases identified as plausible off-targets were then followed up with dose-response experiments to ascertain the potency against each kinase.
- Line 342: “sdosing” should be “dosing”.
- We thank the reviewer for highlighting our typographical mistake. We have corrected this.
Reviewer 2 Report
Comments and Suggestions for Authors
As is well known that introducing the fluorine atom onto molecules have the effect of greatly changing the physico-chemical and biological properties of organic molecules. H. W. Ong and his group described for the improvement of the pharmacokinetics for a potent inhibitor of CSNK2 which includes the substitution of the aryl-hydrogen on the previous parent compound to a fluorine atom.
Their paper is well organized in terms of the contents and includes enough references for cover. I think the manuscript is enough for publication on Molecule in the present format, but considering the context, it is probably more suitable for other journals such as Pharmaceutics or Pharmaceuticals. Small comments are below.
1. In the co-crystal structure, the fluorine atoms did not show any interaction with the kinase. Is the result related to the sustained inhibition of CSNK2 catalytic activity or SARS-CoV-2 replication in vivo?
2. Why did compound 6 attack the ortho-position of the nitro group of compound 5 to give product 7? Please describe the reason for the selectivity.
Author Response
- In the co-crystal structure, the fluorine atoms did not show any interaction with the kinase. Is the result related to the sustained inhibition of CSNK2 catalytic activity or SARS-CoV-2 replication in vivo?
- Indeed, we did not observe a direct interaction of the fluorine atom with CSNK2, as shown by the X-ray structure. However, we showed that addition of the fluorine atom resulted in a compound with reduced the propensity for metabolic clearance in vivo, and hence improved the in vivo exposure of compound 2.
- Why did compound 6 attack the ortho-position of the nitro group of compound 5 to give product 7? Please describe the reason for the selectivity.
- This was an unexpected product but was an experimental result we have obtained. We have added the 1H-19F HOESY spectra and analysis of compound 7 to the supplemental information to provide additional evidence for the structure (Figure S2).
Round 2
Reviewer 1 Report
Comments and Suggestions for Authors
I appreciate the authors' responses to my concerns arising from their original manuscript, and recommend the publication of the revised manuscript in Molecules.